## Overview Review

mental health; community-based mental health; primary level workers; treatment and care gap; India; review

**Corresponding author:**
Mukta Gundi;
Email: mukta.gundi@apu.edu.in

# Diversity in approaches in community-based mental health interventions in India: A narrative review and synthesis

Mukta Gundi[1] ⓘ, Rhea Kaikobad[1] ⓘ and Seema Sharma[2]

[1]School of Development, Azim Premji University, Bengaluru, India and [2]School of Development, Azim Premji University, Bhopal, India

## Abstract

Mental health is a global priority, fundamental to the health and development of all nations. The contribution of mental disorders to the global burden of disease is widely recognized; however, a significant care gap exists, particularly in the context of low-and middle-income countries. In India, for instance, there are 0.3 psychiatrists per 1,00,000 population. To address this severe shortage of mental health professionals and resources globally, the World Health Organization has suggested the adoption of a community-based mental health care approach, where the locus of services shifts from institutional care to local communities. Over the last five decades in India, diverse approaches to mental health care have emerged because of the interaction of dominant discourses on community-based mental health care with various socio-cultural contexts. In addition to the government-run mental health program and programs run by medical colleges, civil society organizations have increasingly contributed to this space. Although studies have assessed individual interventions, there exists a need to map these interventions and synthesize the approaches for service delivery to inform public health practice in India and in low-and middle-income countries at large. This narrative review attempts to map and synthesize insights from community-based mental health interventions in India implemented across diverse contexts. We searched peer-reviewed journal articles and book chapters published in the English language between 2010 and 2023. We present the synthesis of approaches used in 41 community-based mental health interventions, where we unpack key intervention components and processes adopted for primary prevention and promotion; identification and case detection; treatment and care, and rehabilitation in the community. This review presents key recommendations for practitioners about the role of community, the diversity and commonalities in various approaches across contexts, the roles of various actors in service delivery, and the shared values guiding the conceptualization and implementation of community-based mental health interventions in India.

## Impact statement

Mental health conditions are highly prevalent globally. Mental health disorders enormously burden individuals and their families, diminish quality of life and reduce life expectancy. Thus, high-quality mental health care is essential for the development of all nations. While the contribution of mental disorders to global disease burden has been widely recognized, a significant care gap exists, especially in low-and middle-income countries such as India. To address the severe shortage of mental health professionals and resources, the World Health Organization has suggested the adoption of a community-based mental health care approach where the locus of services is shifted from institutional care to local communities. Our review focuses on community-based mental health interventions across community and health care platforms in India, and highlights the adoption of diverse approaches, encompassing primary prevention and promotion; identification and case detection; treatment and care; and rehabilitation. These interventions use a public health paradigm that goes beyond 'diagnosis and illness' to 'wellbeing and recovery' by addressing social determinants affecting mental health through collaborative and intersectoral action. Our narrative review provides key insights to public health practitioners working in the community-based mental health space. First, the idea of a community moves beyond a passive platform for service delivery to an active sociocultural space with valuable community knowledge that shapes the approaches used in these interventions. Second, task-sharing can be reconceptualized beyond the objective of filling the treatment gap, with primary level workers serving as context-experts or experts-by-experience. Third, there are many possible ways in which innovative approaches suggested by the Lancet Commission on Global Mental Health and Sustainable Development can be implemented in resource-poor contexts. Finally, our synthesis underscores the value of practice-based knowledge on community-based mental health in providing insights for public health practitioners across the globe.

## Introduction

Mental health is a global concern. Mental disorders are among the top 10 leading causes of ill-health worldwide and cause 125.3 million Disability Adjusted Life Years (DALYs), according to the Global Burden of Disease Study 2019 (Santomauro et al., 2021). Moreover, since the COVID-19 pandemic, the prevalence of mental health conditions has risen sharply, leading to a significant rise in depressive and anxiety disorders (Santomauro et al., 2021). Thus, high-quality mental health care is a global priority. A critical challenge that most countries face in this regard is the colossal mental health care gap, which is particularly pronounced in Low- and Middle-Income Countries (LMICs) such as India, where less than 2% of the health budget is spent on mental health and over 75% of the people with mental health conditions receive no treatment or care (Murthy, 2017; World Health Organization, 2022). In LMICs, mental health services are largely concentrated near major cities or in psychiatric hospitals, and there is a severe shortage of mental health professionals (World Health Organization, 2022). In India, for example, there are only 0.3 psychiatrists per 1,00,000 population (Gururaj et al., 2016). The adoption of community-based mental health care (CMH) has been seen as a possible solution to address these pressing issues, with the locus of treatment being shifted from psychiatric hospitals to communities (World Health Organization, 2022).

The concept of CMH is intricately linked to the push for deinstitutionalization that took place in high-income countries (HICs) around the 1950s (Burns, 2014; Patel et al., 2018; Thornicroft and Tansella, 2013). The decline of the mental hospital or asylum as the method of treating mental health conditions was spurred by concerns around human rights abuses, quality of care, social exclusion and cost-effectiveness of treatment at long-stay psychiatric institutions (Lamb and Bachrach, 2001; Lord et al., 2001; World Health Organization, 1975). The proposed alternative was to reorganize mental health service delivery from long-stay institutions to the community- where the community is seen as a geographical location and a place of intervention in which people with mental health conditions can be located and treated outside of psychiatric hospitals (Balagopal and Kapanee, 2019a; Jain and Jadhav, 2008; Lamb and Bachrach, 2001; Patel et al., 2018; Thornicroft et al., 2010; Thornicroft and Tansella, 2013). This continues to be the dominant idea driving CMH, which is defined by the World Health Organization (WHO) as "*any mental health care provided outside of a psychiatric hospital*" (p-189) (World Health Organization, 2022).

To provide mental health care in the community, the WHO recommended the strategy of integrating mental health services with primary health care, and training community health volunteers or non-specialist health workers to provide basic mental health care (Jain and Jadhav, 2008). Initial experiments around deinstitutionalization in India were limited to identifying cases in the community and delivering biomedical services (Jain and Jadhav, 2008). The creation of the National Mental Health Program in 1982 shifted the epidemiological focus to a wider public health approach, as it highlighted the need for community engagement and for extending the reach of mental health services by training community health volunteers. This approach, later known as 'task-sharing,' is championed by the WHO through its mhGAP initiative, and is also a key facet of India's state-run District Mental Health Program (DMHP) (Jain and Jadhav, 2008; Patel et al., 2018; World Health Organization, 2010).

Population orientation, ensuring individuals' rights to autonomy and consent, recovery-oriented services, recognizing the role of social determinants of mental health, and evidence-based practice are the guiding principles of CMH (Thornicroft et al., 2016; World Health Organization, 2021a, 2021b, 2022). Contemporary global discourse on CMH is shaped by the movement for Global Mental Health (GMH), which links mental health to sustainable development and emphasizes the need to bridge treatment and quality of care gaps, especially in LMICs (Patel et al., 2018). The existence of mental health along a continuum, across the life-course was recognized by the GMH, thus emphasizing a need to move beyond the biomedical approach in mental health care (World Health Organization, 2022). This evolution in the global mental health discourse called for a wide network of context-specific mental health services spanning prevention, promotion, treatment, and rehabilitation across health and non-health settings, implemented by both formal and informal actors (Patel et al., 2018; Thornicroft et al., 2016; World Health Organization, 2022). Partly as a response to this call, India saw the emergence of CMH interventions with diverse approaches. Apart from the government-run DMHP and community psychiatry programs run by medical colleges, the last two decades have witnessed the rise of CMH interventions implemented by non-governmental organizations (NGOs) in diverse contexts and in multiple forms (Thara and Patel, 2010; Visalakshi et al., 2023).

Previous scholarship has acknowledged that "*mental health is a global public good and is relevant to sustainable development in all countries, regardless of their socioeconomic status, because all countries can be thought of as developing countries in the context of mental health*" p. 1553 (Patel et al., 2018). For this reason, mapping and synthesizing the approaches used in CMH interventions in LMICs such as India can provide learning for practitioners working in the CMH space. Several studies have described or assessed the impacts of some of these interventions (Balagopal and Kapanee, 2019b; Balaji et al., 2012; Joag et al., 2020a; Patel et al., 2010). However, very few studies bring forth holistic and synthesized insights from different models of CMH interventions in India, which can contribute to evidence-informed community-based practice in mental health (Srinivasan et al., 2023; van Ginneken et al., 2017). We attempt to bridge this gap by (i) identifying CMH interventions in different contexts in India and, (ii) unpacking key components of CMH practice, in order to provide recommendations that can inform CMH practice in diverse contexts in India and worldwide.

## Methods

Literature suggests that although innovative approaches have been tried out in low-resource settings, they have often been excluded from reviews due to the lack of generalizable impact evaluations (such as randomized control trials) that are often mandated by the international community as inclusion criteria (Orr and Jain, 2015). The interaction of factors such as social context, organizational characteristics, and local knowledge shapes innovative practices. Randomized control trials may not be able to capture such factors effectively to isolate generalizable and replicable causes of change (Srinivasan et al., 2023). Acknowledging this, we decided to adopt a broad canvas by referring to literature based on diverse types of study designs, including qualitative and mixed-methods studies in addition to impact evaluations. This was in line with the objective of this narrative review, which attempts to map and synthesize the approaches used in existing CMH interventions in India, rather than to evaluate and assess the quality of the evidence.

As a first step, we decided to cast a wide net by searching for peer-reviewed journal articles and book chapters on CMH interventions (which comply with the WHO definition of CMH) published in the English language between the years 2010 and 2023 (World Health Organization, 2022). We conducted the primary search on PubMed, JSTOR, and Google Scholar. We also examined references-of-references as well as physical references in the University library. Some examples of the search words used (in differing combinations) were "community," "mental health," "community mental health," "community-based mental health care," "community mental health program" "intervention," "community-based rehabilitation," "psychosocial rehabilitation," "India."

As a second step, we listed these articles by each intervention, where articles using different methodologies to describe or study the same CMH intervention were pooled together. We identified 177 articles that described or studied a total of 103 CMH interventions from different contexts in India. As a third step, we charted these 103 interventions on an Excel sheet based on indicators such as name of the organization, name of the intervention, geographical location, setting (i.e. urban, rural or both urban and rural), target population and key aspects of program approaches and components. Next, we used the following criteria for inclusion of CMH interventions in the final basket: the intervention should have a) completed at least 1 year of implementation anywhere in the Indian context, and b) some components of direct service delivery involving Primary Level Workers (PLWs) from the community. PLWs include Lay Health Workers (LHWs), who are from the local community with little or no specialized formal training in mental health; Primary Health Professionals (PHPs), who are health care professionals with no mental health specialization; and Community Professionals (CPs), who are professionals working outside the health sector in community settings (van Ginneken et al., 2021). While establishing the criteria of PLWs being 'from the community', we understood the community

not just in terms of location, but also in the wider sense of common factors that shape the identity of a group of individuals, including culture, shared practices and values, common experiences, or characteristics such as health diagnosis (Burgess and Mathias, 2017). We included interventions irrespective of the type of mental health condition being addressed, including interventions responding to general distress among community members. We excluded interventions specifically focused on children with mental health conditions. To decide whether an intervention met our inclusion criteria, when required, we referred to grey literature (see Figure 1). This process led to the final basket of 41 CMH interventions considered in this review.

Finally, we synthesized (see Supplementary Appendix 1) the intervention components by their implementation approaches, by referring to the World Bank Disease Control Priorities for Mental, Neurological and Substance-use Disorders classification: 1) primary prevention and promotion, 2) identification and case detection; and 3) treatment, care and rehabilitation (Patel et al., 2016; Petersen et al., 2016; Shidhaye et al., 2016a).

We further synthesized the intervention components by implementation platforms *i.e. 'the level of the health or welfare system at which interventions can be appropriately, effectively, and efficiently delivered'* (p. 201), to categorize them into community platforms (such as neighborhoods, community groups) or health care platforms such as Primary Health Centers (PHCs) based on - 'where the intervention is delivered' and 'who is providing the services' (Petersen et al., 2016; Shidhaye et al., 2016a). This helped in identifying the locations and the actors for different intervention components. To capture the first line of mental health care provided at community platforms, we added it as a separate platform category under 'treatment and care approaches'. We also treated rehabilitation as a separate category to capture the different community-based rehabilitation approaches used in the interventions included in this

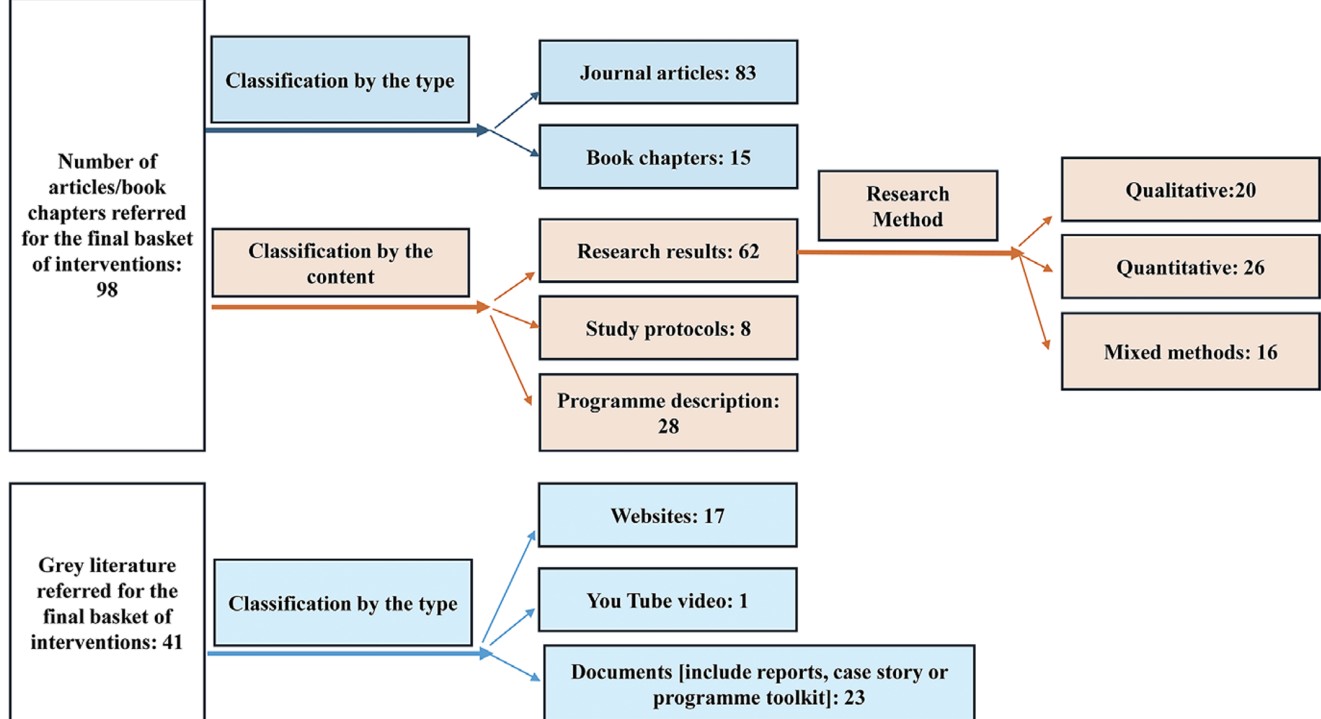

**Figure 1.** Sources of evidence that informed the synthesis of approaches used in the final basket of 41 CMH interventions.

study. This synthesis was done based on the identified peer-reviewed literature and grey literature (when needed) for the final basket of interventions.

## Findings

### Profile of the included interventions

Tables 1 and 2, respectively, provide the profile of the included interventions and the characteristics of each intervention with a tick-chart for the approaches used by their implementation platforms. As shown in Table 2, about half of the (20 out of 41) interventions included had components covering all four approaches (on at least one platform) viz. primary prevention and promotion, identification and case detection, treatment and care, and rehabilitation.

Figure 2 shows the regional inequality in the implementation sites of CMH interventions. Most of these interventions ($n = 31$) were implemented in a single state, while a few ($n = 10$) had their presence in multiple states. Fourteen single-state interventions were from southern states and seven from western states in India. Although multi-state interventions were implemented across different regions in India, a substantial proportion of these were primarily found in the southern, eastern, and western states as compared to central, north-eastern, and northern states.

### Primary prevention and promotion approach as a part of CMH interventions

Although the concept of primary prevention was earlier used only in the context of infectious diseases, the Commission on Chronic Illness in 1957 mentioned primary, secondary, and tertiary prevention in the context of non-communicable diseases (Singh et al.,

**Table 1.** Profile of the CMH interventions

| Intervention characteristics | Number of interventions |
|---|---|
| 1. **Intervention setting** | |
| a. Rural only | 22 |
| b. Urban only | 3 |
| c. Both rural and urban | 16 |
| 2. **Implementation site/s:** | |
| a. In a single state | 31 |
| b. In multiple states | 10 |
| 3. **Implementing institutions for the included interventions** | |
| a. NGOs (independently or in collaboration with the government, academic institutions or with partner organizations) | 30 |
| b. General hospitals (Independently or in collaboration with international network of charity) | 5 |
| c. Independent medical research institute | 1 |
| d. Psychiatric institutions (with at least partial state funding) | 3 |
| e. Government | 2 |

2022). Primary prevention efforts target the general population or individuals at risk, whereas promotion efforts include initiatives to empower people to take control of their own mental health and wellbeing, rather than focusing on mental illness alone (Singh et al., 2022). Twenty-six (out of the 41) CMH interventions had primary prevention or promotion components delivered on community platforms.

### Primary prevention activities focused on raising awareness about mental health on community platforms

Twenty-four CMH interventions (i1-i3, i7-i9, i11, i13, i15, i19, i21, i22, i24, i28, i29, i31, i33-i38, i40, i41) conducted awareness activities for primary prevention which were largely focused on raising awareness, addressing misconceptions and providing information about locally available services to community members and influential stakeholders (such as village heads, municipal authorities, local health personnel, schoolteachers, social workers, police and traditional healers). Awareness sessions for such local leaders served multiple functions. These included sensitization on mental health issues (i3, i15, i34), reducing resistance to access mental health services (i11), and enabling referrals of people with mental illness (PwMI) by providing information about common symptoms (i1, i13, i19, i28, i34, i35, i37).

Traditional healers emerged as important stakeholders in this regard, as they were often the first point of contact for community members facing mental health conditions in different contexts. In three interventions (i13, i34, i37), awareness programs empowered faith-healers to sensitize communities about mental illness and encourage referrals to biomedical clinics. Many awareness programs with community members (i1, i8, i22, i31, i34, i40) focused on debunking myths around mental illness, alleviating stigma, and providing information about the need for regular follow-up and treatment adherence. Fewer awareness programs focused on the social determinants of mental health. Articles on two interventions (i2, i34) specifically described awareness activities with community members that addressed social determinants such as alcoholism, domestic violence, unemployment and armed conflict.

The locations for community awareness activities varied widely across interventions. These included home visits (i11, i21, i40), existing gatherings for general health education in the community (i8), and monthly meetings of Self-Help Groups (SHG), farmer's groups or women's cooperatives (i22, i19, i40). Several interventions also conducted large-scale meetings in public spaces such as street corners, local schools, clubs and public healthcare facilities (i13, i21, i22, i24, i29, i37). Three interventions (i31, i35, i40) used a mobile van that screened films to reach various areas.

Films emerged as a particularly significant tool in generating awareness, with nine interventions using this modality to convey culturally relevant messages about mental health in regional languages. For example, a multi-state intervention (i2) equipped LHWs with short films that depicted scenarios of common issues, such as alcoholism and domestic violence, to initiate discussions on mental health and inspire potential solutions. Two other interventions used short clips from popular cinema to raise awareness about mental disorders (i31, i40). Films were also used as part of campaigns to dispel misinformation around mental health and its determinants. For example, one intervention (i38) showed the community members short videos of PwMI talking about their experiences of illness and recovery, as part of a stigma-reduction campaign. Another intervention (i35) addressed environmental determinants by screening a film to dispel rumors and panic in tsunami-affected areas in Tamil Nadu. Apart from films, other

**Table 2.** Characteristics of CMH interventions and the approaches used

| S no | Intervention name | Primary implementing institution | Setting | Approaches and implementation platforms | | | | | |
|---|---|---|---|---|---|---|---|---|---|
| | | | | Primary prevention and health promotion at community platforms | Identification and case detection at community platforms | Identification and case detection at health care platforms | First line of mental health care at community platforms | Treatment and care at healthcare platforms | Rehabilitation at community platforms |
| i1 | **Asia Psychosocial Rehabilitation Programme** <br> Cohen et al. (2011) | Dr. Somerville Memorial CSI Medical College and Hospital | Rural | ✓ | ✓ | | ✓ | | ✓ |
| i2 | **Atmiyata** <br> Centre for Mental Health Law and Policy (2024), Joag et al. (2020a, 2020b), Pathare et al. (2023), Shields-Zeeman et al. (2017), and World Health Organization, (2021a) | Centre for Mental Health Law and Policy (CMHLP) | Rural and Urban | ✓ | ✓ | | ✓ | | ✓ |
| i3 | **BasicNeeds Model of Mental Health and Development** <br> Basic Needs India (2020) and Underhill et al. (2017) | BasicNeeds India (in collaboration with partner NGOs) | Rural and Urban | ✓ | ✓ | | ✓ | | ✓ |
| i4 | **Community Based Rehabilitation (CBR), Jagaluru** <br> James et al. (2019, 2022), Sivakumar et al. (2019, 2020, 2022) | National Institute of Mental Health and Neurosciences (NIMHANS) and Association for People with Disability (NGO partner) | Rural | | ✓ | ✓ | ✓ | ✓ | ✓ |
| i5 | **Community Care for People with Schizophrenia in India (COPSI)** <br> Balaji et al. (2012) and Chatterjee et al. (2011, 2014) | Sangath, Schizophrenia Research Foundation (SCARF), Chennai Parivartan and Nirmittee (in collaboration with academic institutions) | Rural and Urban | | ✓ | ✓ | ✓ | ✓ | ✓ |
| i6 | **Community Interventions in Psychotic Disorders (CoInPsyD)** <br> Kumar et al. (2017, 2016) | National Institute of Mental Health and Neurosciences (NIMHANS) | Rural | | ✓ | ✓ | ✓ | ✓ | |
| i7 | **Community mental health programme** <br> Ashadeep (2017, 2023), Ashadeep: A Mental Health Society (n.d.), and Narzary et al. (2019) | Ashadeep | Rural and Urban | ✓ | ✓ | ✓ | ✓ | ✓ | |
| i8 | **Community Mental Health Programme (CMHP)** <br> Balagopal and Kapanee (2019c) and Nimgaonkar and Menon ((2015) | Association for Health Welfare in the Nilgiris (ASHWINI) | Rural | ✓ | ✓ | ✓ | ✓ | ✓ | ✓ |
| i9 | **Community Mental Health Programme (CMHP)** <br> Mindlis et al., (2015), Shah et al. (2020), The Mental Health Innovation Network (n.d.), and The MINDS Foundation (2021, 2022) | MINDS Foundation | Rural | ✓ | ✓ | | ✓ | ✓ | ✓ |
| i10 | **Community Mental Health Programme (CMHP)** <br> Balagopal and Kapanee, (2019a; Srinivasan *et al.* Srinivasan et al., 2023) | Mental Health Action Trust (MHAT) | Rural and Urban | | ✓ | ✓ | ✓ | ✓ | ✓ |
| i11 | **Community outreach programme** <br> Giri et al. (2021) | Ranchi Institute of Neuropsychiatry and Allied Sciences (RINPAS) | Rural | ✓ | ✓ | | ✓ | | |

(Continued)

**Table 2.** (*Continued*)

| S no | Intervention name | Primary implementing institution | Setting | Approaches and implementation platforms | | | | | |
|---|---|---|---|---|---|---|---|---|---|
| | | | | Primary prevention and health promotion at community platforms | Identification and case detection at community platforms | Identification and case detection at health care platforms | First line of mental health care at community platforms | Treatment and care at healthcare platforms | Rehabilitation at community platforms |
| i12 | **Dance Movement Therapy for Trauma Recovery** <br> Chakraborty, (2020), Chakraborty and Sen (2019), and Kolkata Sanved (2022, 2023) | Kolkata Sanved | Rural and Urban | | | | ✓ | | ✓ |
| i13 | **Dava Dua** <br> Basu, (2014; Saha et al., 2021; Shields et al., 2016) | The Altruist | Rural | ✓ | ✓ | ✓ | | ✓ | |
| i14 | **Depression in Late Life (DIL)** <br> Dias et al. (2017, 2019), and Reynolds et al. (2017) | Sangath | Rural and Urban | | ✓ | ✓ | ✓ | | ✓ |
| i15 | **District Mental Health Programme (DMHP), Karnataka** <br> Basavaraju et al. (2022), Ibrahim et al. (2021), Manjunatha et al. (2018, 2019), Manjunatha et al. (2021), Parthasarathy et al. (2021), and Rahul et al. (2021) | National Mental Health Programme, State Department of Health and Family Welfare, Karnataka | Rural and Urban | ✓ | ✓ | ✓ | ✓ | ✓ | ✓ |
| i16 | **Enhanced care by community health workers** <br> Pradeep et al. (2014) | St. John's Medical College Hospital | Rural | | ✓ | ✓ | ✓ | ✓ | |
| i17 | **Healthier Options through Empowerment (HOPE)** <br> Bansal et al. (2021), Fathima et al. (2023), and Srinivasan et al. (2018) | St. John's Medical College Hospital | Rural | | ✓ | ✓ | ✓ | ✓ | ✓ |
| i18 | **Home Again** <br> Narasimhan et al. (2019), Padmakar et al. (2020), The Banyan (2024), and World Health Organization (2021a) | The Banyan | Rural and Urban | **Not specified (NS)** | | ✓ | ✓ | ✓ | ✓ |
| i19 | **Integrating mental health in a PHC** <br> Prashanth et al. (2017) | Karuna Trust (in collaboration with Government of Karnataka) | Rural and Urban | ✓ | ✓ | ✓ | ✓ | ✓ | ✓ |
| i20 | **Integration of mental health into Self-Help Groups** <br> Rao et al. (2011) | Sampark | Rural | | ✓ | | ✓ | | |
| i21 | **Janamanas** <br> Balagopal and Kapanee (2019b) and Mukherjee (2021) | Anjali-Mental Health Rights Organization | Rural and Urban | ✓ | ✓ | | ✓ | ✓ | ✓ |
| i22 | **Maanasi** <br> Jayaram et al. (2011) and Jayaram et al. (2019) | St. John's Academy of Health Sciences in collaboration with Rotary Bangalore Midtown and Rotary Howard West | Rural | ✓ | ✓ | ✓ | ✓ | ✓ | ✓ |
| i23 | **MANAS** <br> Buttorff et al. (2012), Patel et al. (2010, 2011), and Pereira et al. (2011) | Sangath | Rural and Urban | | | ✓ | | ✓ | ✓ |

(*Continued*)

| S no | Intervention name | Primary implementing institution | Setting | Approaches and implementation platforms | | | | | |
|---|---|---|---|---|---|---|---|---|---|
| | | | | Primary prevention and health promotion at community platforms | Identification and case detection at community platforms | Identification and case detection at health care platforms | First line of mental health care at community platforms | Treatment and care at healthcare platforms | Rehabilitation at community platforms |
| i24 | **Mental Health Care and Research Foundation (MEHAC)**<br><br>Sunder et al. (2021), Venkateswaran et al. (2014), and Venkateswaran and Vincent (2018) | Mental Health Care and Research Foundation (MEHAC) | Rural | ✓ | ✓ | ✓ | ✓ | ✓ | ✓ |
| i25 | **Mental Illness Treatment Alliance (MITA)**<br><br>Kaul (2022) and The ANT (The action northeast trust) (2022, 2024) | The action north-east trust (the ant) | Rural | | ✓ | | ✓ | | |
| i26 | **Multi-pronged psychosocial intervention for people with mental health and epilepsy problems**<br><br>Mathias et al. (2018a, 2020) | Burans | Rural and Urban | | ✓ | | ✓ | | ✓ |
| i27 | **Nae Disha**<br><br>Kermode et al. (2021), Mathias et al. (2018b, 2019) | Burans | Rural and Urban | ✓ | ✓ | | ✓ | | ✓ |
| i28 | **NALAM**<br><br>Balagopal and Kapanee (2019d), 'Centre for Social Needs and Livelihood (CSNL): NALAM - The Banyan' (n.d.), Mental Health Innovation Network (n.d.), Mukherjee (2021), and Narasimhan et al. (2019) | The Banyan | Rural and Urban | ✓ | ✓ | ✓ | ✓ | ✓ | ✓ |
| i29 | **Naya Daur**<br><br>Bhattacharya et al. (2021), Chatterjee and Roy (2017), Srinivasan et al. (2023), and World Health Organization (2021a) | Iswar Sankalpa | Urban | ✓ | | ✓ | | ✓ | ✓ |
| i30 | **Participatory Learning and Action (PLA)**9<br><br>Ekjut, (2022), Rath et al. (2010, 2020), and Tripathy et al. (2010) | Ekjut | Rural | ✓ | | ✓ | | ✓ | ✓ |
| i31 | **Programme for Improving Mental Health Care (PRIME)**<br><br>Lund et al. (2012), Mendenhall et al. (2014), Shidhaye (2015), and Shidhaye et al. (2016b, 2019a, 2019b) | Sangath | Rural | ✓ | ✓ | ✓ | ✓ | ✓ | |
| i32 | **Project Shifa**<br><br>Ebenezer and Drake (2018) | Padhar Hospital | Rural | | | ✓ | | ✓ | |
| i33 | **Psychosocial support in tsunami-affected areas**<br><br>Padmavati et al. (2020) and World Health Organization (2006) | Schizophrenia Research Foundation (SCARF) | Rural | ✓ | | ✓ | | ✓ | ✓ |

**Table 2.** (*Continued*)

| S no | Intervention name | Primary implementing institution | Setting | Approaches and implementation platforms | | | | | |
|---|---|---|---|---|---|---|---|---|---|
| | | | | Primary prevention and health promotion at community platforms | Identification and case detection at community platforms | Identification and case detection at health care platforms | First line of mental health care at community platforms | Treatment and care at healthcare platforms | Rehabilitation at community platforms |
| i34 | **SAWAB Intervention** <br><br> Malla et al. (2019b, 2019a) | Supporting Always Wholeheartedly All Broken Hearted (SAWAB) (in collaboration with an academic institution) | Rural | ✓ | ✓ | ✓ | ✓ | ✓ | ✓ |
| i35 | **SCARF Telepsychiatry in Puddukottai (STEP)** <br><br> Tharoor and Thara (2020) | Schizophrenia Research Foundation (SCARF) | Rural | ✓ | | ✓ | | ✓ | ✓ |
| i36 | **Seher Urban Community Mental Health and Inclusion Programme** <br><br> Davar (2012) and Davar et al. (2021) | Bapu Trust | Urban | ✓ | ✓ | | ✓ | ✓ | ✓ |
| i37 | **SHIFA** <br><br> Burgess and Mathias (2017) | Emmanuel Hospital Association | Rural | ✓ | ✓ | ✓ | ✓ | ✓ | ✓ |
| i38 | **Systematic Medical Appraisal, Referral and Treatment (SMART)** <br><br> Maulik et al. (2015), Maulik et al. (2017b), Maulik et al. (2017a), and Tewari et al. (2021), The George Institute for Global Health India (2024) | The George Institute for Global Health | Rural | ✓ | | ✓ | | ✓ | |
| i39 | **Thinking Healthy Programme (THP))** <br><br> Fuhr et al. (2019) | Sangath | Urban | | | ✓ | ✓ | | |
| i40 | **Vidarbha Stress and Health Program (VISHRAM)** <br><br> Shidhaye et al. (2017) | Sangath | Rural | ✓ | ✓ | ✓ | ✓ | ✓ | |
| i41 | **Yuva Spandana** <br><br> Banandur et al. (2021), Banandur et al. (2020), and Garady et al. (2021) | Department of Youth Empowerment and Sports, Government of Karnataka | Rural and Urban | ✓ | | | | | |

a. White cells with a '✓' indicate presence of an approach.
b. Blue dotted cells with a '✓' indicate intervention approaches are implemented where the boundary between the community platforms and health care platforms are blurred.
c. Grey shaded cells indicate an absence of the approach in the intervention.

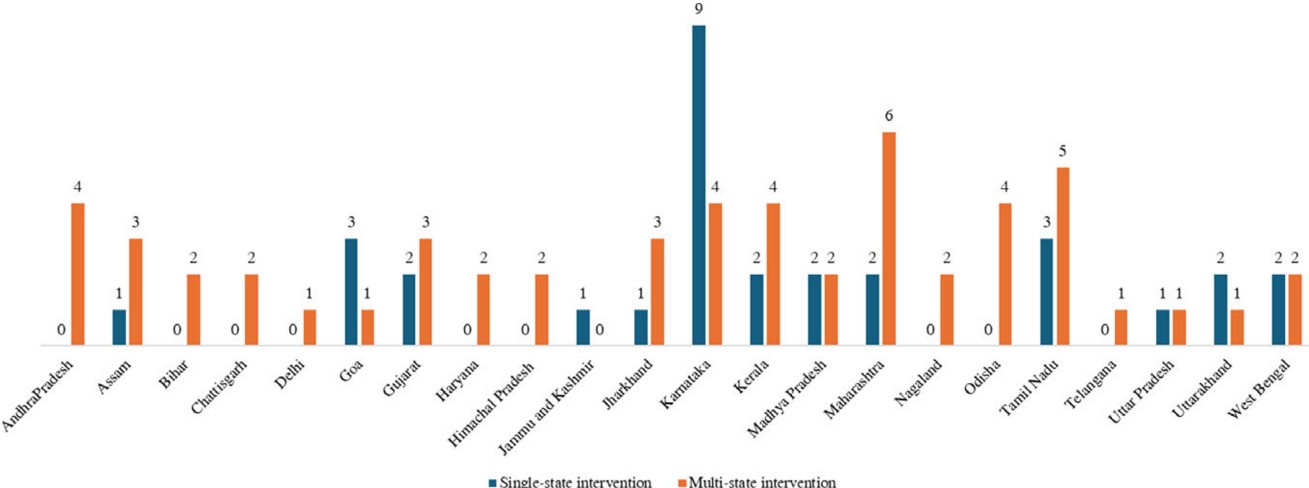

**Figure 2.** States with presence of CMH interventions included in this review.

modalities such as theatre, music, visual art and writing were also used to generate awareness, through activities such as street plays (i8, i22, i35, i38), broadcasting drama and songs on mental health during TV and radio programs (i34), display of flip charts, posters and wall paintings (i8, i15, i38), and distribution of brochures and leaflets (i9, i21, i38).

### *Mental health promotion activities focused on enhancing capacities to foster wellbeing on community platforms*

Seven interventions (i1, i15, i27, i28, i30, i37, i41) focused on mental health promotion, through sessions aimed at building capacities for mental health and well-being. Such activities were often led by LHWs (including peers) and generally took place in groups, with only one intervention (i41) providing one-on-one services at specially established youth guidance centers. Mental health promotion components of the interventions aimed to enhance emotional resilience (by working on topics such as self-care, stress management and problem solving), to build skills for interpersonal communication, and to help navigate common stressors related to education, career, gender and sexuality (i1, i27, i37, i41). Two interventions also focused on collectively developing solutions to commonly faced problems in the community, through participatory activities such as games, storytelling and role-play. The first was a Participatory Learning and Action (PLA) intervention in Jharkhand and Odisha (i30), which brought community members together to identify, prioritize and ideate responses to issues such as perinatal depression, gender-based violence and severe mental illness. The second was an Uttarakhand-based youth mental health promotion program (i27), which mobilized participants to implement a project addressing social determinants of mental health in their community.

Twenty-five (out of 26) CMH interventions with a component of primary prevention and promotion also engaged in identification and case detection in the community, potentially indicating how such programs can enhance early identification of people with mental health conditions, prompting early referrals for mental health care and treatment.

### *Identification and case detection approaches as part of CMH interventions*

Evidence from both LMICs and HICs suggests programs that train PLWs to identify PwMI and detect cases on community platforms as a 'good practice'. This is likely to help in early intervention, thus, reducing long-term treatment costs (Petersen et al., 2016). Literature indicates that case detection and diagnosis of more complex mental health conditions are carried out on health care platforms, with both PLWs and specialists playing a role in the detection and referral process (Shidhaye et al., 2016a; World Health Organization, 2022). Our synthesis found 39 interventions (except i12, i41) that had components of identification and case detection.

### *Interventions with components of identification and case detection on community platforms*

In 26 interventions (i2, i4-i10, i13-i17, i19-i22, i24, i26-i28, i31, i34, i36, i37, i40), community platforms were used for identification and case detection. This was primarily done by LHWs through house-to-house visits (e.g. i16, i24, i27, i37, i40) or through engagement with local stakeholders such as Accredited Social Health Activists (ASHAs) (*e.g.* i4, i15, i17, i24, i26, i31), community members or leaders (e.g. i2, i7, i10, i26, i34) or through community-based health screening fairs (e.g. i17, i20). A few interventions (e.g., i16, i36, i37) trained LHWs in using screening instruments or surveys to identify people with mental health conditions. Interventions in Gujarat, Karnataka and Jammu and Kashmir (i13, i19, i34) engaged with faith healers who were trained in actively identifying and referring people with mental health conditions for psychiatric and psychological treatment.

### *Interventions with components of identification and case detection on health care platforms*

Identification and case detection were carried out on health care platforms in 23 interventions (i4-i8, i10, i13-i19, i22-i24, i28, i31, i34, i37, i39, i40). This primarily took the form of diagnosis done by PHPs (i8, i16, i17, i22, i23, i40) or by psychiatrists (i4-i7, i10, i13, i28, i31, i34, i37) at PHCs, CHCs or clinics. PLWs conducted screening at health care platforms in three interventions (i14, i23, i39). A multi-state intervention (i18), which aimed at providing non-institutional living arrangements for people admitted to psychiatric institutions, implemented a unique approach to identify PwMI who could not be re-integrated into their families. In this intervention, identification was done either at the organization's psychiatric facility or at selected mental health hospitals run by the government.

### Interventions with components of identification and case detection where the boundary between community and health care platforms is blurred

Our synthesis showed that in 10 interventions, case detection and diagnosis were done at locations where the boundaries between community and health care platforms seemed to blur. These included- camps set up in the community (e.g. i1, i3, i11, i25, i32, i38) and telepsychiatry provided at a central location in the community or in a mobile van (e.g. i30, i33, i35). In an intervention in West Bengal (i29), a hybrid platform was created at street corners where homeless PwMI lived. Trained PLWs engaged with community members to identify such homeless individuals so that a psychiatrist could provide a diagnosis by visiting these localities.

It is important to note that all CMH interventions with a component of identification and case detection provided a referral linkage to community-based first line of care, community-based treatment camps, private clinics or nearby government facilities. We discuss this in greater detail as our next finding.

### Mental health care and treatment approaches in CMH interventions

The WHO has recommended a tiered approach to meet the treatment and care-related needs of people with mental health conditions (World Health Organization, 2022). Such an approach encompasses complimentary components provided by diverse actors comprising the first line of mental health care provided by trained PLWs (community platforms) along with primary and specialist services (health care platforms) (Petersen et al., 2016; Shidhaye et al., 2016a; World Health Organization, 2022).

### First line of mental health care at community platforms

A total of 38 interventions provided the first line of mental health care at community platforms (i1-i12, i14-i22, i24-i40), in which trained PLWs (such as community volunteers, ASHAs or peers) played a key role.

Thirteen interventions used innovative psychosocial approaches implemented by PLWs, which were delivered either at the individual level or in groups. Interventions offered at individual level included manualized counseling using techniques such as active listening, problem-solving and behavioral activation (i2, i14, i39, i40); home visits to collaboratively formulate and implement personalized recovery plans (i5, i18, i26, i30, i36); and kiosk-based counselling to address restrictive gender norms (i21). Five interventions providing such care used group-based participatory mental health activities (i12, i26, i20, i33, i36) to create non-judgmental spaces to discuss problems, share coping strategies and for collective grieving (i17, i20, i33). Group sessions often involved arts-based components such as story-based resources to build reflective conversations (i26), arts-based therapy (i36), and dance movement therapy (DMT) (i12).

Along with this, in 20 interventions (i1, i3, i4, i6-i11, i15-i17, i19, i24, i28, i31, i32, i34, i35, i38), LHWs provided follow-up services (such as psychoeducation, reminders on appointments, and medication adherence) to complement ongoing biomedical care. In an intervention implemented in Andhra Pradesh and Haryana (i38), a digital app was used to provide ASHAs with instructions for conducting follow-up visits with PwMI.

Our synthesis suggests that government-appointed LHWs (ASHAs) had a role in providing mental health care only in a few CMH interventions (e.g. i4, i15, i17, i38) despite their cadre already being present in the community. Most of the interventions hired LHWs or trained volunteers from the community to provide the first line of mental health care on community platforms.

### Treatment and care at healthcare platforms

Approaches to treatment and care provided at health care platforms were multilayered and involved diverse actors. We found a few interventions that provided psychological or psychosocial treatment at health care platforms, delivered either by PLWs (*via* manualized counselling) (i23, i31, i40) or by specialists (i10, i13, i24, i34, i37).

The primary focus of the treatment provided at health care platforms was found to be biomedical in nature. This involved the integration of mental health into primary health services through diagnosis and prescription of medication by general physicians (i8, i15, i16, i17, i19, i22, i23, i31, i38, i40), often under the guidance of a psychiatrist. Several interventions had visiting psychiatrists who themselves engaged in diagnosis and treatment (i4, i5, i6, i7, i10, i13, i15, i18, i24, i25, i28, i34, i37, i40), at PHCs, Community Health Centers (CHCs) or private clinics.

It was evident from our synthesis that 14 CMH interventions (i3, i4, i6, i7, i15, i16, i17, i19, i23, i28, i31, i37, i38, i40) primarily used the existing government spaces such as PHCs, and CHCs as the main healthcare platform to provide biomedical treatment. A few interventions referred people with severe symptoms to their own general or psychiatric hospitals for further treatment (i1, i8, i11, i18, i22, i28, i32). Some interventions provided referrals to partner hospitals or organizations for further treatment (i19, i21, i36). For example, an intervention in urban Maharashtra (i36) provided treatment and care support on health care platforms through active partnerships with general physicians, psychiatrists, and Ayurveda, Yoga, Naturopathy, Unani, Siddha, and Homeopathy (AYUSH) practitioners.

### Treatment and care where the boundary between community and health care platforms is blurred

Our analysis brought forth a few examples where the boundary between community and health care platforms seemed to blur, as treatment was provided either through camps, where temporary health care platforms were created in community settings (i1, i11, i25, i32, i33, i38), or through telepsychiatry, where a virtual health care platform was created close to the community (i30, i35), to provide specialist care. An intervention supporting homeless individuals with psychosocial disabilities in West Bengal (i29) engaged PLWs who provided psychoeducation and psychosocial support to homeless PwMI upon their consent. The boundary between community and health care platforms was blurred as this intervention involved a psychiatrist regularly visiting street corners to provide diagnosis and biomedical treatment to those in need.

### Community-based psychosocial rehabilitation approaches in CMH interventions

Community-based psychosocial rehabilitation aims to support people with mental health conditions to achieve their optimal functioning and inclusion in the community (World Health Organization, 2022). This essential component in CMH interventions helps to build competencies among people with mental health conditions and addresses diverse social factors that affect people's mental health, so that they can live productive, satisfying, and dignified lives within their communities (World Health Organization, 2022).

Twenty-eight CMH interventions (i1-i5, i8-i10, i12, i14, i15, i17-i19, i21-i24, i26-i30, i33-i37) had components of rehabilitation, which broadly focused on enhancing competencies of people with mental health conditions to cope with day-to-day living; providing linkages to welfare services; in-kind services and support; enhancing access to livelihood opportunities; and creating a more supportive environment by strengthening relationships between people with mental health conditions and their communities (e.g. potential employers). Across all settings, LHWs played a key role in implementing such components.

### Rehabilitation components on community platforms to address livelihood-related needs

The need for livelihoods emerged as a significant issue, with 13 interventions (i3, i5, i8, i12, i15, i18, i19, i22, i24, i28, i29, i34, i35) enabling people with mental health conditions to access employment opportunities. This was facilitated in multiple ways. Three interventions (i5, i15, i24) were focused on equipping PwMI with basic vocational skills. One (i12) aimed at training survivors of gender-based violence as DMT practitioners. Three interventions (i5, i22, i28) provided access to employment opportunities through referrals to vocational training courses run by other organizations. Four interventions (i3, i8, i18, i29) supported and guided PwMI by understanding how they would like to be employed and motivating them to join (or re-join) work. Seven interventions (i3, i5, i8, i19, i22, i29, i35) actively networked with potential employers such as NGOs, local vendors, daycare centers and nurseries to address the stigma around hiring people with mental health conditions.

Apart from facilitating access to employment opportunities, other strategies were adopted to help individuals with mental health conditions to supplement their income and gain financial independence. Eleven interventions (i2, i4, i5, i14, i18, i19, i21, i28, i29, i35, i37) enabled access to livelihoods by facilitating linkages to government welfare schemes. This entailed networking with local authorities to create documents such as ration cards, job cards (under schemes such as Mahatma Gandhi National Rural Employment Guarantee Act) and disability certificates, which would entitle recipients to an additional set of welfare schemes. Four interventions (i1, i3, i26, i35) supported the establishment of SHGs, so that people with mental health conditions could access financial resources. Three interventions (i8, i28, i30) also provided materials (such as livestock, seeds and tailoring machines) through which participants could begin to generate their own income. Two interventions (i10, i29) also adopted a more direct, charitable approach by collecting donations of food and clothing from community members.

### Rehabilitation components on community platforms to enhance the competencies to navigate daily stressors

Various strategies including individualized support given by LHWs at home (i5) or in residences set up by the intervention (i18), training at daycare centers (i10) and group activities at accessible locations in the community (i17, i30), were used to enhance the competencies of PwMI to navigate daily stressors. Four interventions (i5, i10, i15, i17) used cognitive and behavioral methods, including improving functioning on Activities of Daily Living (ADL), social skills training, recreation, problem-solving skills, and relaxation exercises (i5, i10, i17). One intervention (i30) conducted PLA meetings in which LHWs focused on building skills to address social barriers to mental wellbeing.

### Rehabilitation components on community platforms to address shelter and housing needs

To address shelter or housing needs, three interventions provided daycare services (i10, i24, i15) or ran halfway homes (i15) for PwMI. One intervention in Tamil Nadu (i33) provided guidance and practical support in obtaining housing for tsunami survivors. A multi-state intervention (i18) implemented a unique approach to community-based supported living by establishing long-term accommodations outside institutional settings, in which PwMI opted to stay with a group of peers.

### Rehabilitation components on community platforms to promote family and social inclusion

Strengthening social support networks and enhancing social inclusion are key to rehabilitation, with 18 interventions (i1, i3-i5, i9, i17-i19, i21, i26-i30, i34-i37) involving activities addressing these issues. Ten interventions (i3, i5, i9, i17, i18, i19, i26, i27, i28, i30) created peer support networks for PwMI and their families. One intervention (i37) used peer support meetings as an avenue to strengthen the collective voice of people with mental health conditions and caregivers, so that they could advocate for their own rights. Two interventions (i21, i27) enabled people with mental health conditions to navigate crises such as gender-based violence through linkages to legal and community support structures.

Apart from such direct support, several interventions also worked with families and communities to promote inclusion. Four interventions (i4, i5, i34, i36) equipped family caregivers with knowledge about supporting PwMI. Two interventions (i1, i30) promoted community inclusion by facilitating the participation of people with mental health conditions in social activities (*e.g.*, SHGs, PLA meetings) involving people from the wider community. Two other interventions mobilized neighbors and local community members to come together as a safe, responsive and caring support system for PwMI who were homeless (i29) or facing crises, such as violence in the family (i36).

## Discussion

This narrative review provides a synthesis of 41 CMH interventions implemented in various parts of India. Our work highlights a staged approach to mental health and wellbeing, involving a range of actors providing services on community and health care platforms, for a wide spectrum of mental health conditions, beyond the binaries of presence or absence of disease (Patel et al., 2018).

We showcase diverse intervention approaches spanning from primary prevention, identification and case detection, treatment, and care to rehabilitation (Patel et al., 2018). Twenty out of 41 included interventions adopt all four approaches (Table 2 and Supplementary Appendix 1) in a staged manner to cater to the mental health needs of communities. Our synthesis unearthed that community platforms (primarily in the neighborhood, at the doorstep or at community groups) were utilized to implement all four approaches, whereas health care platforms were primarily used for identification and case detection, and for treatment and care. Our findings underscore the important role played by LHWs in bridging the gap between community and health care platforms, thereby ensuring continuity of care.

In the light of the alarming scarcity of resources for mental health care services in India, efforts to deinstitutionalize mental health have led to the emergence of community 'as a platform' (Kohrt et al., 2018; Shidhaye, 2015). However, our review uncovered the diversity in the

conceptualization of what constitutes 'community' in the CMH space, beyond just a platform for service delivery. The idea of 'community' ranged from the population in focus for the intervention (such as PwMI, people facing general distress), PLWs (such as LHWs, PHPs), local stakeholders (such as local leaders, teachers and faith healers), to general community members engaged in creating an inclusive space for recovery. CMH interventions addressed the mental health needs among diverse populations, including individuals sharing a social identity and a geographical space such as people living in a village with shared history (i8, i2) and homeless individuals in an urban area (i29); individuals sharing similar health experiences or diagnoses such as people with schizophrenia (i5), people at the risk of suicide (i40) or pregnant women (i30); individuals sharing difficulties or past trauma such as sexual violence (i12), natural disaster (i33) and armed conflict (i34); or people with a shared values for seeking mental health care such as faith-based healing (i13).

Our findings corroborate the body of evidence that recognizes the key role played by LHWs in closing the treatment gap (Kohrt et al., 2018; Patel et al., 2018; Raviola et al., 2019; Shidhaye, 2015). Beyond this, LHWs from the community provide contextual and experiential expertise while actively working within community settings. Trained LHWs play a significant role in providing peer support and in activating social networks for PwMI, which is essential for creating favorable social conditions to seek and sustain mental health treatment and care (Patel et al., 2018). Active involvement of local stakeholders in a supportive role, and engagement of general community members to destigmatize mental health, also contributes to strengthening such networks and creating inclusive spaces. Therefore, meaningful participation of LHWs and community members broadens the idea of 'community' from a passive site of intervention and simply a platform for service 'delivery,' to an active sociocultural space with valuable community knowledge shaping CMH approaches (Bayetti et al., 2023; Orr and Jain, 2015).

Apart from LHWs, a crucial role is also played by other actors and public health practitioners in the CMH space. Interventions included in our review involved diverse actors such as people with relevant professional competencies in social work and mental health (e.g. i1, i2, i29, i31), PHPs such as general physicians who provide biomedical treatment at PHCs (e.g. i8, i19, i31), and people with professional degrees in mental health who can train PLWs (e.g. i8, i22) and provide specialist treatment when needed (e.g. i1, i15, i30, i35, i40). These diverse actors bring together biomedical, psychological and social-work related competencies and perspectives, shaping CMH as a plural space for mental health practice (Patel et al., 2018).

Our findings also show different ways in which included interventions are utilizing the four innovations suggested by The Lancet Commission on Global Mental Health and Sustainable Development to be scaled up (Patel et al., 2018). The first innovation, which consists of task-sharing with LHWs, with reference to psychosocial interventions, was reflected in several interventions which had rehabilitation components (e.g. i2, i5, i8, i14, i17, i18, i21, i22, i27, i28, i29, i30, i34, i35). Some interventions providing treatment and care engaged LHWs in providing psychosocial support through manualized counselling and therapeutic groups (e.g. i2, i17, i26, i40). LHWs in such interventions contribute either as part of the stepped care approach (e.g. i17, i23, i31, i33, i40) or as primary providers (e.g. i2, i12, i14, i21, i26) (Barnett et al., 2018; Patel et al., 2018).

The second innovation suggests that LHWs coordinate with primary and specialist care in resource-constrained settings. This has been implemented in certain interventions where mental health treatment was provided in the existing government (e.g. i4, i6, i15, i19, i31, i37, i38, i40) or in non-government (e.g. i5, i8, i18, i22) health facilities, where LHWs played a key role in coordination and referral to ensure continuity of care. However, efforts to improve the capabilities of primary care staff were found to be limited, as more interventions relied on psychiatrists conducting regular visits (at government health facilities), camps or telepsychiatry consultations for diagnosis and treatment, with fewer general physicians delivering mental health treatment.

The third innovation suggested by the Lancet Commission relates to the adoption of digital platforms to facilitate the delivery of interventions across the continuum of care. This was seen in a few interventions and used in various ways such as telepsychiatry to deliver mental health treatment (e.g. i30, i33, i35), the use of an app to coordinate treatment (e.g. i25, i38), and the display of films on mobile phones (e.g. i2, i35) to spread awareness about mental health. However, none of the included interventions seemed to use digital tools across the continuum of care. There is a need for further research on the scope, benefits and challenges of using digital tools across the continuum in the Indian context.

The fourth innovation involves implementing community-based interventions to enhance help-seeking and demand for care (Patel et al., 2018). All the included interventions that had a component of primary prevention incorporated this innovative approach through awareness sessions and efforts to destigmatize mental health among community members.

Our narrative review also found a fifth type of innovation that can be adopted in the form of arts-based approaches for mental health care and promotion. Modalities such as films, street plays, visuals and music (i2, i8, i22, i34, i35, i38) have been used to implement such approaches in culturally appropriate ways. In addition, participatory arts-based sessions involving storytelling, role-plays, music and DMT (i12, i20, i26, i27, i30, i33) have been used to create a non-stigmatizing space for mental health promotion and care, where communities can actively prioritize, have reflective conversations, and ideate potential solutions around mental health concerns. It has been found that such arts-based innovative approaches push for a holistic recovery by addressing social determinants of mental health by enhancing social cohesion, socioemotional skills, community capital, inclusion and various aspects of mental wellbeing especially among people from vulnerable backgrounds (Fancourt and Finn, 2019). However, there is a need for more evidence to understand the scope to scale up such approaches in the Indian CMH space.

Our work highlights the key role played by NGOs as implementing institutions and stakeholders in the CMH space in India. Previous scholarship from LMICs has discussed how NGOs, with their social aim and community embeddedness, play a significant role in shifting the biomedical paradigm of mental health to a more inclusive social paradigm through innovation, impact, scale, and sustainability (Bayetti et al., 2023; Kohrt et al., 2018; Mathias et al., 2024; Srinivasan et al., 2023). This is notable, especially as many included interventions were initiated outside of formal health systems, though some programs have collaborated with health systems for a specific approach. Literature suggests that NGO services are inequitably distributed across different settings in India, with metropolitan areas having more mental health care provided by NGOs, as compared to rural areas (Mathias et al., 2024). However, our findings deviate from this, as we found more CMH interventions in rural areas as compared to urban areas. This can be partially explained by our specific criteria focusing on interventions involving PLWs. As rural areas are less resourced in

terms of mental health services and specialists as compared to urban areas, our search criteria possibly prompted more interventions from rural areas that relied on task-sharing by PLWs (World Health Organization, 2022).

Several interventions began as a response to the needs of individuals in distress and trauma (i12, i35), or to the devastating impact caused by natural disasters (i33), or as an extension of palliative care (i10, i24), thus underscoring the shared value of prioritizing human life across interventions. By upholding the rights and dignity of PwMI, by asserting the idea of collective responsibility among communities, and by respecting historical and cultural perspectives, CMH approaches endorse public health philosopher Dan Beauchamp's argument that "*public health should be a way of doing justice, a way of asserting the value and priority of all human life*" (p. 6) (Beauchamp, 1976).

Interventions with components of psychosocial rehabilitation explicitly prioritize improving quality of life through intersectoral action, emphasizing the value of recovery and reintegration into community and not merely being symptom-free (Patel et al., 2018; World Health Organization, 2022). These efforts in some interventions, thus embrace an expansive idea of deinstitutionalization that upholds the values of recovery-oriented care, recognizes the role of social determinants, addresses complex needs relating to mental health, and focuses on community inclusion and participation, while also bridging the treatment gap (Patel et al., 2018; Thornicroft et al., 2016; World Health Organization, 2022). Our synthesis indicates certain gaps in CMH approaches implemented by the included interventions. First, we did not come across more than one intervention that specifically addresses the mental health needs of the elderly (i14), despite existing evidence on the increasing mental disorders and distress among the elderly in LMICs (Dias et al., 2019). Second, we did not find any CMH interventions implemented at workplaces offering mental health prevention, promotion and care within occupational health and safety programs. This misses a potential opportunity to train human resource managers, employers and trade union representatives with mental health support skills (Petersen et al., 2016). Third, the integration of mental health into general health care was found to be limited. For instance, integration of mental health specifically into health programs that address tuberculosis, HIV, or NCDs was found only in one intervention (i17). Therefore, far more efforts need to be made to ensure that existing resources are utilized to provide holistic health for individuals in need (World Health Organization, 2022).

We acknowledge the limitations of our work. Despite keeping a broad canvas for inclusion, some CMH interventions may not have found a place in this review due to their resource constraints to be able to publish a peer-reviewed article or a book chapter. As this review did not adopt the systematic review methodology, we cannot fully negate selection bias. Additionally, the approaches of certain interventions were available in a fragmented manner in the referred peer-reviewed literature. This may have certain implications on synthesis; however, to address this limitation, we referred to grey literature wherever necessary. Although the framework chosen by us to synthesize the intervention approaches helped categorize them into primary prevention and promotion, identification and case detection, treatment and care, and rehabilitation, the boundaries between these categories were fluid in certain cases. For example, interventions aimed at creating a safe space or inclusive environment for PwMI achieved more than what 'rehabilitation' as a conceptual category would encompass, as efforts to create an inclusive space may also contribute to promoting mental health in

the community. Our ability to explain such interlinkages may have been affected by the availability of details and component descriptions in the referred literature. Additionally, the included literature for the State-led CMH intervention (i15) is location-specific. Our scope does not extend to other geographies where this intervention is implemented, and we also acknowledge the critique of this intervention for having heavily adopted a pharmacological approach, with fewer efforts to address social inequalities (Jain and Jadhav, 2009; Shidhaye et al., 2019b; Srinivasan et al., 2023).

Despite these limitations, to the best of our knowledge, this is the first attempt to map and synthesize the literature on CMH intervention approaches from diverse contexts in India. Our strength lies in an inclusive approach to methodology that includes the embedded knowledge generated from public health practice, which often does not find space in reviews (Orr and Jain, 2015). Lastly, our review has the potential to inform CMH practice globally, especially in the LMICs facing a wide mental health treatment gap.

## Conclusion

Our narrative review presents critical reflections that can inform comprehensive, community-based, recovery-oriented practice, using the framework provided by the World Bank Disease Control Priorities (Petersen et al., 2016; Shidhaye et al., 2016a). Our review suggests that by treating 'community' as an active sociocultural space with valuable community knowledge, and by improving the capabilities of PLWs to provide mental health treatment and care, resource-poor contexts across the globe can be reimagined as 'human resource-rich contexts'. Several examples in this review suggest reconceptualizing the idea of task-sharing beyond the objective of filling the treatment gap, as trained LHWs working with different types of implementing institutions can serve as 'context-experts' and 'experts-by-experience' (Bayetti et al., 2023; Orr and Jain, 2015). As half of the global population lives in countries with acute scarcity of mental health specialists, these examples from India provide a broad canvas to imagine how implementing institutions, beyond the State, can address the mental health needs of communities using innovative approaches, as suggested by the Lancet Commission (Patel et al., 2018; World Health Organization, 2022).

These insights are based on embedded knowledge from CMH practice across different contexts in India. As the Lancet Commission has recognized that both LMICs and HICs have resource-poor contexts, these insights will help public health practitioners across the globe, enabling them to respond to mental health needs of communities in order to avoid suffering, loss of human capabilities and human life (Orr and Jain, 2015; Patel et al., 2018).

**Open peer review.** To view the open peer review materials for this article, please visit http://doi.org/10.1017/gmh.2025.10046.

**Supplementary material.** The supplementary material for this article can be found at http://doi.org/10.1017/gmh.2025.10046.

**Acknowledgements.** We thank Prof. Arima Mishra and Prof. Edward Pinto for their inputs as peers during the initial process of intervention search. We thank Prof. Adithya Pradyumna for his input on the methodology draft. We are grateful to the two anonymous reviewers for their valuable feedback.

**Author contribution.** Mukta Gundi and Seema Sharma contributed to conceptualizing this review. All authors have contributed to the synthesis of the articles and interpretation. Mukta Gundi and Rhea Kaikobad have written the manuscript. All authors have contributed to the review of the manuscript, have read and approved the final manuscript.

**Financial support.** This study was funded by Azim Premji University under the Faculty Small Research Grants by School of Development (Project fund code: UNIV RC00347). This study is one part of the grant received for the project titled "*Community mental health interventions: Learnings for research and practice*" that entailed scoping review and primary research. The views expressed are those of the authors and do not necessarily those of the funders. The funders had no role in study design, data analysis and synthesis, decision to publish, or preparation of the manuscript.

**Competing interests.** We have no known conflicts of interest to disclose.

**Ethics statement.** This work did not involve any primary data collection.

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
