## [Reviewer Report]

This is an interesting paper reflecting important work done in communities across India to address the mental health treatment gap.

I recommend that you try and shorten it as it is currently far over the word count limit and a long paper to read through. I recommend putting some of your data into tables/figures instead of reporting it all, or adding supplementary data. I have also suggested a different way of reporting the findings which should reduce word count.

In your discussion, I’d also like to see a bit more of a summary of your findings and implication for community mental health service provision in India. i.e., what are the implications of your review, and what are the recommendations or key points you would like to make, given everything you have found from your review. As it stands it is almost a waste of all of your hard work not using this as a platform to make some conclusions and recommendations from all that you have found out about community mental health services across India. Including what is predominant, and what is lacking – what needs further attention and resources to improve MH services in India.

Below are more specific comments:

LINE 77 onwards – You don’t need to say ‘henceforth’, you can just give the acronym.

Paragraph on line 148 – is it necessary here to state what each reviewer did? I thought that was stated in ‘author contributions’ at the end?

170 – regarding your Inclusion criteria, as far as I can see there were only 2 inclusion criteria – 1 that they should be implemented longer than 1 year, and 2 that they involved PLWs from the community.

What about the following factors:

Type of mental disorder / classification of MD?

Type of treatment included? (e.g. was there a ‘baseline’ for what constituted as mental health treatment?)

(You could have members of the general population receiving mental health promotion, or diagnosed patients receiving yoga, which, if all, do you include?)

193 – please add a section or two describing how you are going to use the word ‘platform’. It is confusing at times, for example, in line 366 & 7 you say health care platform and then ‘community platform’, yet your review is supposed to be about ‘community platforms’ as an umbrella term if I am correct.

201 - I am wondering why you do not have table 1 as a table describing the characteristics of all of your studies. E.g. authors/implementing NGO, intervention type, area, interventionists, rural/urban, target population, key aspects of programme etc. I would highly recommend that you add this.

Line 203 – what is your rationale for splitting studies up by rural, urban, and a combination, as you describe them throughout the findings? If there isn’t a rationale for splitting them up this way then I would recommend reporting them only by treatment approach. (ie primary prevention and promotion; identification and detection, and Mental health care and treatment …?) (as you describe in the introduction to your discussion). This would help to shorten the findings section as, although it is interesting, it is far too long at the moment. (I would be interested to know how it got submitted given the word count limitation in the journal?)

In addition, is there a way of putting some of the detail you report in the findings in tables instead, to reduce word count?

213 - Table 1 – I would like to see added to Table 1 – type of interventionist / facilitators; type of intervention (prevention / treatment etc),

Line 301 – you write: “Evidence from both LMICs and high-income countries highlights the effectiveness of programs that train PLWs to identify PwMI and detect cases in the community” - Why / how are detection programmes effective, and in what?

Line 313 – I recommend using normal brackets ( ) rather than square brackets [ ] throughout your text.

Line 337 – See above comment - what do you mean by detection “on community platforms”? Please clarify.

Then the three references for the studies, I think you can just use I i39, i29, i36, rather than quoting the references again, as you have already referenced these three studies in section 2. If you have a characteristics table, with study numbers, then you wouldn’t have to repeatedly list all the study authors when you list them under each section. This applies throughout your paper.

(line 338 - why is the WHO reference in there?)

344 – again why 3 references for this one study? If there was a characteristics table this might be clearer for the reader.

Lines 392-395 – see above comment about study references.

Please provide a clearer distinction between the paragraphs starting:

“386 In rural settings (21 out of 22) interventions provided first line of mental health care at

community platforms.”

And 403 “Twenty-one (of 22) interventions implemented in rural areas had a component of providing treatment and care at health care platforms.”

Line 462 – check your writing for unnecessary repetition/descriptions – e.g. in the section under rural and urban, you don’t need to say “Four interventions implemented in both rural and urban settings provided referrals…”

Discussion

In the discussion I would suggest also referencing interventions by their numbers, not their references.

582 – suggest the word ‘staged’ not staging approach. Please say more about this ‘staged’ approach and what it means for CMH?

598 - To wrap up this paragraph – what are the implications of your findings that there is a wide range of conceptualisations of what ‘community’ constitutes? What does this mean for CMH? (It is fine to have a summary of your findings, but then you need to link it to other literature and summarise what this means for the provision of mental health services in India.

599 – do you mean needs of ‘individuals’ in distress and trauma?

607 – I would make this a separate paragraph.

608- I would say “some” interventions attempt to embrace… (not all of them do)

613 – the implementation of which approach? Not sure what you are referring to?

614 – are you summarising the findings of your review? If so, then say, “interventions included in this review involved people with” … relevant professional competencies in… xxx”

Or, “This review found practitioners who were… xxx”

633 – remove ‘the’

640 – change ‘of’ to ‘for’

645 – I would change the word ‘suggests’ to ‘found’

648 – add the word ‘and’ between task sharing and use of digital platforms

654 – Could you make it clearer how art-based approaches / intersectoral approaches address social determinants of mental health?

665 – remove the full stop

662 – add ‘’the’ Mental Health care act…

664 – it would be interesting to have a sentence or two discussing the reliance on NGOs to provide mental health care, due to the fact that the state care is inadequate? It seems a blatant admission that state care is inadequate if they include the necessity of NGOs in their MHCA policy?

668 – “This can be partially explained by our specific criteria focusing on interventions involving PLWs. As it has been well-acknowledged that rural areas are less resourced than urban areas, which has possibly prompted more interventions with a task- sharing approach in rural areas as compared to the urban areas”

I suggest re-writing this as it doesn’t make sense – I am not sure what your selection criteria has to do with task-sharing in rural areas?

700 – you write that “Our review presents critical reflections on these approaches that can inform a comprehensive, community-based, recovery-oriented practice to address mental health-related needs globally.”

If this is the case, what are the reflections that inform a “comprehensive, community-based, recovery-oriented practice to address mental health-related needs globally.” ? I can’t see these. A summary of these would form a good basis for your conclusion.

---

## [Reviewer Report]

Overall Comments:

This paper represents a potentially valuable contribution to understanding of community based mental health interventions in India. Key strengths of the paper include its contextual background, use of the WB Disease Control Priorities for MNS classification system as a means of organizing the findings, clear writing style and consistent use of terminology.

The authors provide a sound rationale for their choice of methodological approach, electing to do a narrative review that enables a general synthesis and description of diverse CMBH interventions rather than attempting a systematic review. This seems a sensible approach given the heterogeneity of interventions and evaluation approaches employed. The introductory section situates the rise of CMBH interventions in the context of global mental health developments concisely, which sets the stage for the papers aims. The authors also do a good job of introducing and defining the various terminology used throughout the paper (e.g. community, PLWs, CHM) and employing the terms consistently without falling prey to overuse of acronyms, which makes it simple to follow the subsequent discussion with ease.

Another important strength of the paper is its use of the WB Disease control priorities for MNS classification system as a means of organizing the study findings and identifying subcomponents. However, the paper would benefit from a more systematic presentation of the descriptive findings as well as further attempts to highlight trends and gaps in intervention components and to link these to global trends. For this reason, I have suggested major edits.

Detailed Comments:

The discussion section is divided based on a slightly modified version of three classifications presented in the WB Disease Control Priorities for MNS. This is a sound choice and serves to increase the global relevance of the article by providing an analytical frame that is of potentially universal interest. However, I do not find any clear rationale for the authors decision to further subdivide the description of intervention components and findings based on the classifications of urban, rural or both. Indeed, the existence of a large category of interventions which cover both urban and rural areas renders this distinction somewhat moot. While it’s important to collect this information as part of the descriptive data and discuss its implications (as the authors do towards the end of the article) the utility of focusing on this distinction in the analytical section is questionable. I did not come away from reading the article with a clear understanding of whether or how urban, rural and cross-site interventions were substantially different from or similar to one another and feel that focusing on this division actually obscures other relevant trends.

Instead, the paper would benefit from more systematic disaggregation of the diversity of approaches and activities within each of the four main categories. I recommend that the authors try to analyze high level patterns and gaps related to intervention components that can inform future research and intervention work, within the constraints of the narrative review process.

For example, in the section on primary prevention and promotion, it would be helpful to draw attention to the extent to which interventions focus on awareness raising, stigma reduction or demand for services vs. measures that seek to prevent mental illness (e.g. by addressing sociodeterminants) vs. promoting mental wellbeing. It appears that nearly all the interventions target mental health awareness raising activities, while only one attempts to act on the social determinants of health or promote individual capacities to foster wellbeing. This is a potentially important observation for community-based intervention research as it indicates a missed opportunity when it comes to mental health promotion and prevention interventions.

Likewise, for the section on case detection it would be good to be more explicit in describing the extent to which interventions focus on identification and referral vs. diagnosis vs. both activities, and to know where these activities take place (health or non-health locations) and who leads (CHWS, primary care workers, specialists) these efforts. Knowing whether most interventions cover identification, referral and diagnosis components vs only one or the other is relevant when considering intervention design.

The section on treatment and care does a better job of making sense of the descriptive findings, explicitly identifying psychosocial, psychological and biomedical components. Nonetheless, it would be helpful to be more explicit in describing how prevalent each type of care is, whether the different types tend to be delivered by CHW, PHCs, or specialists and to make observations about where the different types of care are provided (health sector, non-health sector, temporary or virtual site).

On rehabilitation, I agree with the authors attempts to break this section out from treatment and care. Notably, the only attempt to address the social determinants of health appears to come at the rehabilitation phase, once individuals have been diagnosed (or at least identified as at risk). This seems to be a shortcoming of all existing intervention models, and an area for further potential innovation.

Response to editorial question: How well do the authors describe how the results fit in with global research and global learnings? And how could this be improved/expanded?:

The introductory section situates the rise of CMBH interventions in the context of global mental health developments in a concise manner. This positions the paper to make a good contribution to global research and learnings. Likewise, the utilization of the WB classification system referenced above enables the findings to be interpreted in a global context and presents a potential template for similar studies in other national, regional or global contexts. However, the lack of a more systematic organization of the descriptive results and limited attempts identify trends or gaps within each classification hampers the authors ability to make links between India specific findings and global or regional trends.

Incorporating a slightly more comparative perspective, for example by discussing the extent to which India can be considered a leader, follower or simply to be “on trend” with respect to the implementation of CMBH models would make for helpful additional context. My impression is that India has been towards the leading edge of adopting community-based interventions and there are some references to the innovative nature of India’s CMBH models in the final section, but the justification for this is absent due to lack of comparison to a global context in the concluding section.

Other Comments:

In addition to eliminating the geographic breakdown within the WB classification categories and related to my encouragement to attempt to identify more high level trends, I think that a quantitative “tick chart” that lists each intervention and its key components would be highly valuable to potential readers. This could be something as simple as simply listing the four classification categories and relevant subcomponents (e.g. for Care and Treatment include: type of treatment, delivery location, type of care provider) on a horizontal axis and the names of interventions on the vertical. I anticipate the readership of this paper will include individuals looking to design or improve interventions and thus interested in the extent to which those reviewed are muti-component, intersectoral etc.

Finally, I think it would be interesting to know which of the interventions reviewed meet the 2018 Lancet Commission criteria referenced in the concluding section (e.g. role of task sharing; coordination with primary and specialist care, adoption of digital platforms, interventions that enhance demand for care (via awareness raising), as this would make this national review even more globally relevant.

---

## [Reviewer Report]

I congratulate you on a thoroughly re-worked and more substantive paper. I would only recommend proof reading the manuscript for grammatical edits of which there are quite a few.

---

## [Reviewer Report]

The authors have clearly taken careful consideration of the reviewer recommendations and substantially revised the article accordingly.

The result is a more concise, readable and useable manuscript that both contributes to scholarly understanding of the nature of community mental health interventions in India, and also has the potential to serve as a valuable reference for researchers and others interested in this topic (e.g. practitioners and decision makers).

Importantly, the reorganization of the findings enables attention to relevant trends and patterns. For example, observations about the important contribution of traditional healers to awareness raising activities, the substantial use of film as a communications medium and the somewhat surprising absence of ASHAs from many first line community platforms were buried in the original manuscript but have now been elevated.

The inclusion of the tick box table is also a very helpful addition. I would suggest that the title of each column be repeated at the start of each page for ease of interpretation (so that the reader does not need to scroll to the top to remember what each tick refers to).

---

## [Editor Report]

I want to thank the authors for making substantial revisions to the manuscript. There are two very minor suggestions from the reviewers: 1. Include the title of each column at the start of each page for ease of interpretation, so that the reader does not need to scroll to the top to remember what each tick refers to. 2. Kindly proof read the manuscript for grammatical edits. 

Kindly address these comments and re-submit. My decision is (almost) accept!!!.